# Attention and Signal Detection

**Adam Reeves**

Department of Psychology, Northeastern University, Boston, MA 02115, USA; reeves@neu.edu

**Abstract:** In this paper, I first review signal detection theory (SDT) approaches to perception, and then discuss why it is thought that SDT theory implies that increasing attention improves performance. Our experiments have shown, however, that this is not necessarily true. Subjects had either focused attention on two of four possible locations in the visual field, or diffused attention to all four locations. The stimuli (offset letters), locations, conditions, and tasks were all known in advance, responses were forced-choice, subjects were properly instructed and motivated, and instructions were always valid—conditions which should optimize signal detection. Relative to diffusing attention, focusing attention indeed benefitted discrimination of forward from backward pointing Es. However, focusing made it *harder* to identify a randomly chosen one of 20 letters. That focusing can either aid or disrupt performance, even when cues are valid and conditions are idealized, is surprising, but it can also be explained by SDT, as shown here. These results warn the experimental researcher not to confuse focusing attention with enhancing performance, and warn the modeler not to assume that SDT is unequivocal.

**Keywords:** signal detection theory; attention; performance

## 1. Introduction

### 1.1. Signal Detection Theory in Perception: A Primer

Signal detection theory (SDT) has been widely used in studies of human visual and auditory perception since its introduction by [1] Green and Swets. A textbook by [2] Macmillan and Creelman provides a nice introduction to what has become a rather developed field. An important aspect of the theory from the point of view of this paper is that it provides a role for a mental concept, namely, attention, which was previously taken to be entirely subjective. Signal detection theory (SDT) provides ways of conceptualizing the role of attention both in processing sensory information and in how decisions about what is sensed are reached. SDT points towards methods of identifying the effects of attention which can also be applied in studies of animals, making attention an objective concept at some cost to its phenomenology.

Intuition suggests that paying more attention should enhance performance; indeed, it has been taken as a defining characteristic in Psychology that items selected by attention will be processed better, and rejected items worse, than neutral items (e.g., Posner et al. [3]). To explain this role for attention in SDT terms requires at least an elementary exposition of SDT, which the reader may skip if this material is already familiar. The essential idea is that stimulation (auditory, visual, tactile) can be represented as activity on a single internal dimension termed 'strength'. Strength represents the evidence accumulated by the senses towards a categorical decision, such as that I see a butterfly—evidence that may rely on one dimension (e.g., sound level) or on numerous dimensions (color, motion, size) which are presumed to be integrated into one evidential signal. The perceiver in an experimental trial is assumed to orient towards a signal or stimulus, denoted by S, and answer 'Yes' when interrogated about its presence if the strength of the evidence (denoted z) elicited by S exceeds a criterion level, denoted c. The perceiver has a forced-choice decision between Yes and No, not being allowed to report 'uncertain'.

Averaged over trials, one can compute the 'hit rate' or probability P (z > c S) that the subject reports 'Yes' given that the signal or stimulus was present, the 'false alarm rate' P (z > c no S), the 'miss rate' P (z < c S) and the 'correct rejection' P (z < c no S). Accuracy, the proportion of correct trials, Pc, is then P (z > c S) Ps + P (z < c no S) (1-Ps), where Ps is the probability of a stimulus being presented. These definitions apply to binary trials, when there are two possible signals or stimulus conditions, and two possible responses. In general, there are four classes of events: a hit {S1, R1}, a false alarm {S1, R2}, a miss {S1, R2}, and a correct rejection {S2, R2}.

These binary definitions apply to 'Yes, No' experiments, in which S2 is null and R2 is 'No', as well as to discrimination experiments in which S1 and S2 are distinct stimuli and R1 and R2 the corresponding identifiers. For example, subjects might discriminate between a square and a triangle, or between a high and a low tone. Since the mathematics are the same, P (z < c S1), P (z < c S2), P (z < c S1) and P (z < c S2) are still termed 'hits', 'false alarms', 'misses', and correct rejections, even though these terms are logical only for the detection experiment. In a search experiment, S1 might be the target (say, an apple) and S2 a set of distractors (a bowl of many other fruits), but again the mathematics are the same.

These four probabilities, P (z > c S) and so on, can be modeled if the internal strength variable is assumed to be corrupted by Gaussian noise, N, with mean 0 and variance 1.0. Other probability distributions are possible, as when light is detected and a Poisson counting variable is required, but only for the Gaussian are the mean and variance mathematically independent, which simplifies matters. Strengths on discrimination trials are either S + N, with mean S, or N, with mean 0, while strengths on discrimination trials are either S1 + N or S2 + N, with means S1, S2. When signals are constant, the corresponding variances all equal 1.0, the noise variance. The hit rate is then Phit = P (z > c S + N), the false alarm rate is Pfa = P (z > c N), and detectability, d′ = z (Phit)-z (Pfa), where z(p) is the inverse normal score, z, associated with probability p, such that z (0.5) = 0, z (0.1) = −1.28, and z (0.9) = + 1.28. The criterion, c = −[z (Phit)−z (Pfa)]/2; this defines c as mathematically independent of d′, i.e., values of one do not constrain the value of the other. A 'conservative' value of c, that is c > 0, implies that the subject reports 'No' more often than Yes; a 'liberal' criterion, c < 0, implies the opposite; and a balanced criterion, c = 0, implies no bias either way (Other definitions of the criterion, such as β, are not independent of d′ and will not be considered further here, although they have their specialized uses [2]).

Note that d′ in an unbiased estimate of S, that is, d′ tends to S, since the mean strength on signal trials is S—much greater than that on noise trials (d′ is an estimate given a finite number of trials; its uncertainty is given in [4]). For example, if Phit = 0.9 and Pfa = 0.1, the detectability score d′ = 1.28−−1.28 = 2.46. As the perceiver makes more hits and fewer false alarms, detectability (d′) increases. Conversely, if Phit = Pfa, so that d′ = 0, the subject operates at chance, reporting 'Yes' equally often whether the stimulus is present or not. Usually, 0 < d′ < 4.0; negative d′s are possible but represent aberrant responding (saying No for Yes and vice versa), whereas d′ > 4 indicates so few erroneous trials as to make estimation suspect.

As the response probabilities are conditional, Ps, the probability of the stimulus, does not enter into the measurement of d′. It therefore becomes an empirical matter whether varying Ps increases, decreases or leaves detectability unchanged. An increase in d′ with Ps may imply an increase in the preparation or 'set' for a stimulus, which is one aspect of paying attention. In 'signal known exactly' experiments, clear versions of S are presented, along with temporal cues to indicate onset and offset of S, so that the perceiver knows what the target is and when it can occur. In contrast, in 'oddity' experiments, the signal is an unexpected deviation from a standard, one whose timing or nature cannot be anticipated. Differences in d′ between signals known exactly and unprepared signals provide evidence that attention to the signal improves sensitivity (e.g., [5] Santhi and Reeves, 2003).

Crime provides a real-world example of SDT. Suppose the world contains dishonest (S1) and honest (S2) individuals, and the response of society is to lock them up (R1) or set them free (R2). Noise (N) is the uncertainty of the evidence, and the criterion (c) is the amount of evidence needed to convict.

Different bodies collect evidence (police) and make Yes/No decisions (jury). Improvements in policing increase d', while changes in laws alter c. The analogy to perception is useful. The sensorium (ear, eye) collects the evidence on which the decision is made (in cortex). Detectability measures the sensory efficiency, and the criterion measures the deciding element. As long as the latter does not interfere with the former, the evidence will not be biased by the desire to obtain one or other answer (as when the judge and jury are segregated from the police to avoid corruption.) Distinguishing a sensory stage from a subsequent decision stage helps to provide a rational account in SDT terms of sensation (the first stage alone), perception (sensation plus a correct decision leading to a hit) and hallucination (sensation plus a wrong decision generating a false alarm or a miss in the case of a negative hallucination).

### 1.2. The Role of Selective Attention

Selective attention is defined here as resources devoted to collecting evidence about a particular target stimulus, S, and rejecting other possible stimuli as noise (N) [6] Fine and Reeves, 2018, tabulate 19 other ways in which attention has been defined and operationalized in psychology and neuroscience). Given a limited time for collection, increasing resources (i.e., increasing attention) will increase d' by increasing signal strength, S; decreasing resources by attending elsewhere will attenuate the stimulus (e.g., Carrasco et al. [7]). This definition includes all-or-none selection, which occur if the stimulus is either selected fully or rejected completely ([8] Palmer, 1995), as well as graded selection (for an example in hearing, see Reeves and Scharf [9]). In the following paper, the term 'focusing' is used for concentrating attention on a stimulus or subset of stimuli, the task set for the subject being to 'monitor' those stimuli. The expected sequence in a selective attention experiment is that, when asked to monitor a subset of stimuli, the subject focuses attention on them, and sensitivity (d') improves.

Altering c does not change the amount of evidence and so is not equivalent to selective attention as defined. Since the overall proportion of correct trials, Pc, can go up based on either better evidence or a more optimal criterion, finding an increase in Pc is not necessarily evidence of more attention, despite claims to the contrary. Careful experimenters often choose an experimental design in which the criterion is balanced (c = 0) in every condition, so that any change in accuracy implies a change in d' and therefore an effect on processing. For example, in a discrimination experiment, S1 and S2 may be presented in successive halves of each trial, and the subject is asked to pick which interval contains S1. So long as any bias towards one or other interval is constant across conditions, changes in accuracy imply changes in d', so such methods are termed 'criterion-free'. Demonstrating an effect of selective attention is easiest if a criterion-free method can be employed; if not, rating methods, not discussed here, may also be used [10] (Egan, 1957.)

We now need one more concept from signal detection theory, that of the signal/noise ratio, S/N. As signal strength diminishes or noise increases, detectability must deteriorate, so d' = S/N. This can be fleshed out by defining the transduction function f(.) of the sensory organ. Signal strength increases monotonically with stimulus energy, E, that is, S = f(E), where f(0) = 0, so that no signal is obtained when the stimulus is off. If S = bE or S = blog(E), for some constant b, the sensory transducer is linear or is Weberian, two cases which (perhaps surprisingly) cover wide ranges of stimulation in hearing, vision, and touch, though obviously do not accommodate color or harmony. The sources of noise (N) are external (environmental), $N_e$, or internal (neural) $N_i$. If both sources are Gaussian, $N = \sqrt{(N_e^2 + N_i^2 - 2rN_eN_i)}$, where r is the correlation between $N_i$ and $N_e$. If external and internal noise sources are independent, r = 0 and $N = \sqrt{(N_e^2 + N_i^2)}$. In this case, $d' = f(E)/\sqrt{(N_e^2 + N_i^2)}$, and $N_i$ can be measured by first removing all external noise, so $N_e = 0$, and then applying various amounts of external noise to find the level which doubles d'; this gives $N_i$ in units of stimulus energy [11], thereby providing a behavioral measure of random neural activity in the sensory transducer.

Given signal detection, that is, d' = S/N, selective attention can increase detectability by increasing S or decreasing N, or both. One way in which attention can decrease noise is by 'noise exclusion', that is, by reducing $N_e$ by filtering out $N_e$ components distant from the signal [12,13]. Attention ($\alpha$) can increase S by increasing the efficiency of transduction [7]. Either S = f($\alpha$E) or S = $\alpha$f(E), depending on

whether attention accesses the sensory input before transduction, as by orienting the eyes for better vision or the head for better sound, or after transduction, as might be in the brain. These mechanisms are not exclusive; attention can both increase S and reduce N.

This ends the primer on elementary signal detection theory. Readers who wish to follow up can read Macmillan and Creelman [2], which is a lucid exposition for experimentalists, or Egan [10] for a deeper mathematical treatment. What is common to every idea mentioned so far is that increasing attention must *improve* performance, either by filtering out noise (e.g., Palmer [8]), or decreasing uncertainty about the signal (Lu and Dosher, [12,13]), or directly enhancing the signal (Carrasco et al. [7]). Foreknowledge of the signal ('attention' to it) permits the subject to improve detectability by creating optimal filters to enhance S, decrease N, or both ([1], Green and Swets, p. 162).

Indeed, evidence that selective attention enhances performance has been often secured. For example, an auditory 'signal' tone presented at an expected frequency is detected more reliably than a 'probe' tone presented at an unexpected frequency ([14] Greenberg and Larkin, 1968; [9] Reeves and Scharf); line segments shown at an expected orientation ('signals') are detected more accurately than line segments ('probes') presented at an unexpected orientation ([15] Kurylo, Reeves, and Scharf); and letters whose locations are cued in advance are reported more accurately than uncued letters ([16] Eriksen and Yeh; [17] Skelton and Eriksen.) Attentional enhancement is not restricted to location in frequency space or visual space; attended objects may be processed more precisely than un-attended ones ([18] Egly, Driver, and Rafal), even when the locations of attended and unattended objects are identical ([19] Blaser et al.).

A focus on signal detection rather than on broader organismic factors is surely justified when subjects are in the same mental state in every experimental condition, and are not distracted, indifferent, lethargic, unclear about the task, overly aroused, or otherwise abnormal. The great advantage of STD in these situations is that the experimenter is required develop methods consistent with the theory and show that d′, not just the criterion, changes with attentive state. In our research ([6], Fine and Reeves), we employed criterion-free behavioral methods so that changes in accuracy would reflect changes in sensitivity, not changes in the criterion.

Critically, the generalization that attention improves performance presupposes *validity*: that is, attention can be paid to the signal rather than to irrelevant or competing locations or features, the task instructions and cues are never misleading, and the stimuli and task are known in advance and well-practiced. Validity permits the subject to process the target signal optimally, i.e., as well as possible given the inevitable noise. Attention to the wrong location (e.g., [14] Greenberg and Larkin) or the wrong spatial frequency band ([20] Yeshurun and Carrasco) or to signals not known exactly [1] causes perceptual decisions to be based on the outputs of less-than-optimal filters, impairing sensitivity.

## 2. Method

The data discussed in this paper are a subset of those presented by Fine and Reeves [6]. The purpose of this paper is to explain the role of signal detection theory in helping understand these data. We [6] had restricted our experiments to those in which *the signal is known in advance, its form and timing are unvarying, and its spatial location is validly cued,* and we had asked whether focusing attention necessarily improves processing even in this restricted domain. We ran conditions in which attention was endogenous by using the 'offset' or placeholder method of Jonides and Yantis [21], as in the exogenous case, attention is by definition facilitatory. Placeholders (boxy numeral 8's) were converted into letters or into optotypes (forward or backward pointing Es) by briefly removing some of the line segments that formed the placeholder in the target location, the remaining three placeholders being unaltered.

Our subjects (a total of 78 young adults, all with 20:20 vision or better) viewed a TV raster screen on which the four placeholders appeared at top, bottom, right, and left of fixation (see [22] for further details). The stimuli (letters, optotypes, and placeholders) were black, presented on a moderately bright white screen. Stimuli subtended 0.57 deg at the eye when shown at 5 deg eccentricity.

After 1025 ms, one of the placeholders changed into a letter for 250 ms (or an optotype for 50 ms) by offsetting the necessary line segments, and then returned to placeholder form for a final 250 ms. The after-coming placeholders were expected to act as backward patterned masks, that is, to restrict processing time [23,24], but were not chosen to make the letters or Es subliminal (and they did not).

Subjects self-initiated trials by pressing the space bar. They reported the direction of the E, forward or backward, or the identity of the letter, by having the subject select (using a mouse) the letter, the E, or the backward E, on a display screen which followed each trial and remained on until the response was collected. Speed and accuracy were recorded for each trial. Conditions were fixed for each block of 24 trials, and subjects were well practiced in each condition beforehand. Critically, the task instructions, to *focus* attention on two locations (left and right, or top and bottom), or to *diffuse* attention to all four locations, were fixed for every block, were flagged to the subject before each block, and were understood by all. Stimulus class (letters or optotypes) was varied across sessions but never within. Thus, every aspect of the experiment was known in advance.

The main result for the 27 subjects of Experiment 1, as reported by us [22], was that when identifying a randomly chosen letter from a set of 20 possible letters, focusing on the top and bottom locations reduced accuracy by10%, and slowed RTs by 49 ms, as compared to diffusing attention to all four locations. These represent large costs for focusing top and bottom. Interestingly, focusing had almost no effect on left and right locations, possibly because left–right reading habits automatized letter identification on the horizontal meridian, or because (as shown for object-based attention) shifts across the horizon are much faster than those in the vertical direction [8] (Barnas and Greenberg [4]). These conclusions were supported by significant interactions in ANOVAs with replications applied both to arc-sine transformed percent corrects ($F_{3,78} = 20.65$, $p < 0.01$) and to reaction times ($F_{3,78} = 7.78$, $p < 0.01$).

We had been concerned that the perverse effect of focusing on top and bottom locations might have somehow resulted from a mid-line hemispheric interaction. However, Experiment 2 in Fine and Reeves [22] presented offset letters in a diagonal format, one in which the 4 placeholders were rotated by 45° to move away from the mid-line. Results showed that, compared to diffusing attention to all four placeholders, focusing on either of the opposite diagonals once again came at a cost, response latency being slowed by 24 ms ($F_{1,14} = 12.5$, $p < 0.01$) and accuracy deteriorating by 4% ($F_{1,14} = 9.91$, $p < 0.01$).

In contrast, in Experiment 3, focusing improved performance with offset optotypes (Es) ($F_{1,19} = 9.2$, $p = 0.007$), by more in the top and bottom locations (by 51 ms) than in the left–right ones (by 7 ms) ($F_{3,57} = 2.71$, $p = 0.049$). Accuracy was high (93%) and did not vary significantly with condition. The improvement in latency is consistent with the usual effects of focusing and in marked contrast to the results with offset letters. Focusing on letters also came at a cost when the response was binary (a two-key choice between 5 vowels or 5 consonants), rather than one out of 20 letters, so the difference between results with letters and those with Es was not due to the type of report (our Experiment 4 in [6].)

## 3. Model

How can we account for the varied effects of focusing attention with signal detection theory? A fundamental comparison is between signals known exactly with signals not known in advance [1]. For example, search for colored disks placed among a few distractors is far better for known than for unknown ('odd') colors [7]. However, in the present case, the stimuli, letter or optotypes, were all known in advance and well-practiced, yet focusing aided the optotypes but hurt the letters. How can attention have opposite effects of this sort, given that a system which can build a filter for an E can also do so for the other letters?

One simple notion is that, in the current conditions, attention *always* improves processing, but can do so more for the noise than for the signals, in which case focusing will lower the signal/noise ratio and hurt performance. In the following development, it is assumed that the costs of focusing on

the top and bottom letter locations cannot be explained by attention to the wrong spatial frequency band [20] (Yeshurun and Carrasco), alluring as that explanation is, because the letters, optotypes, and placeholders were all in the same frequency band and seen at the same eccentricity. It is also assumed that locations separated by 5 deg or more are far enough apart to act as independent channels [25] (Gardner), without mutual interference [26] (Bahcall and Kowler), and can be processed in parallel. Let S/N be the signal to noise ratio, where S stand for signal strength, $\alpha$ for the level of attention, and R for the response, and define $Ne \geq 0$ as the external noise (here due to placeholders) and $Ni > 0$ as the internal noise from random neural activity. In a generic signal-detection model, the total noise, $N = \sqrt{[(Ne)^2 + Ni^2]}$, given that external and internal noise are uncorrelated, so their variances sum. Specializing this generic model to the level of attention, $\alpha$,

$$R = \alpha S / \sqrt{[(\alpha^2 Ne)^2 + Ni^2]} \qquad (1)$$

where S and Ne refer to the transduced signals and noise, and R is proportional to sensitivity measured as d′. The generic model avoids detailing the exact transduction functions that lead from stimulation to S and Ne, assuming only that the signal increases monotonically with stimulus contrast, such that R = 0 when S = 0, and the external noise increases monotonically with the contrast and number of the external noise sources. Focusing is identified here with the attention parameter, $\alpha$, although other ways of varying attention are not precluded. It is critical that $\alpha$ differentially modulates the external stimuli, S and Ne, compared to the internal noise, Ni. If attention modulated all three terms equally, its effects would cancel and the S/N ratio would be invariant to attention. If both S and Ne were modulated by $\alpha$ to the same extent, for example if Equation (1) was replaced by $R = \alpha S / \sqrt{[(\alpha Ne)^2 + Ni^2]}$, then increasing $\alpha$ could only increase R. Only if attention modulates S and Ne unequally, as in Equation (1) where Ne is multiplied by $\alpha^2$, can attention harm as well as benefit.

This generic model predicts sensitivity; its relation to latency is to be specified. It may be pinned down precisely by a diffusion model such as Ratliff's, but here, such a strategy would require a premature specialization. Assuming simply that latency is inverse with response strength, as in the LATER model of [22] (Carpenter et al.),

$$RT = b/R + RTo \qquad (2)$$

Here, b scales 1/R to latency, RT refers to the mean or median reaction time, and RTo to a constant motor component. Equations (1) and (2) together imply that an increase in the S/N ratio improves performance by both increasing sensitivity and speeding the response.

Inspection of Equation (1) shows that when internal noise dominates (Ne << Ni), an increase in $\alpha$ increases S more than N and therefore increases R, so focusing helps, whereas when external noise dominates (Ne >> Ni) an increase in $\alpha$ increases N more than S, so focusing hurts. This is in contrast to signal detection models in which attention can only help (e.g., [8]). These predictions are independent of the form of stimulus transduction, given that attention (in Equation (1)) multiplies the outcome of the transduction, not the stimuli themselves, but a fuller model should also specify the transducer functions.

I fit Equations (1) and (2) to data assuming that focusing on two locations doubled $\alpha$ compared to diffusing attention to all four locations. The internal noise was arbitrarily defined as Ni = 1.0. The model was fit to average speed and accuracy data in [6] jointly, with Ne = 0.1 for optotypes and Ne = 10 for letters, and with RTo = 400 ms and b = 0.17. Figure 1 illustrates the resulting stimulus-response functions, assuming linear transduction for S. Arrows indicate that focusing (by increasing $\alpha$ from 0.5 to 1.0) increases R at all levels of S when Ne = 0.1 (black lines) but decreases R when Ne = 0.1 (grey lines), showing that focusing can have opposite effects depending on the level of external noise. We [22] did not vary S parametrically, so the actual form of the transduction is not known: non-linear transduction would bend the lines into curves, but maintain the same order.

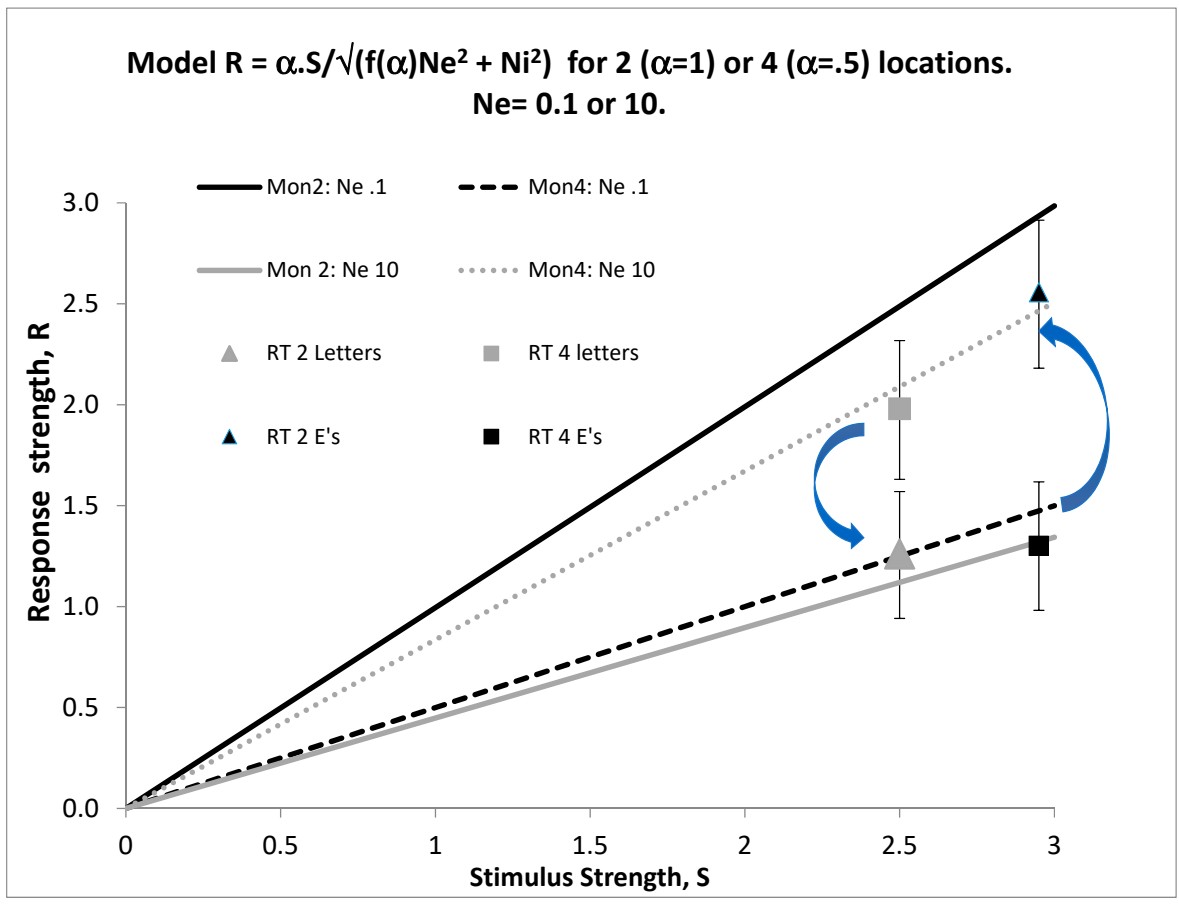

**Figure 1.** Stimulus (S) and response (R), assuming linear transduction for S and constant internal noise (Ni = 1.0). The response is in units of d′. Arrows indicate that focusing improves the response when Ne = 0.1 (black lines) but hurts when Ne = 10 (grey lines). The lines illustrate that focusing can have opposite effects depending on the level of external noise (see text). Data from [6] (Fine and Reeves, 2018) confirm this pattern for specific levels of S. However we did not vary S parametrically, so the form of the transduction is unknown. Squares (RTs in Mon 4) and triangles (RTs in Mon 2) plot latency data converted to rates according to Equation (2) with b = 0.17 and RTo = 0.40 s; grey symbols for letters, black symbols for Es. Bars show +- one standard error around each rate.

One wonders why Ne, or more precisely, Ne/Ni, varied so much between optotypes and letters. Endogenous attention acts as a gate in the model of Reeves and Sperling [24]. If the gate must be held open for longer to identify individual letters than to detect the orientation of the Es, external noise from the placeholders (acting as masks) will be greater in the former than the latter task. That is, holding the gate open for long enough to obtain sufficient information to identify arbitrary letters lets in so much noise from the placeholders that focusing becomes deleterious. An experiment in which the placeholders did not return, so that only forward masking occurred, could test this notion; whereas patterned backward masks limit processing time, forward masks do not [23,27]. If, instead, gating was fixed and Ne was constant, the internal noise (Ni) associated with the Es would have to differ markedly from that for the other letters; given that the Es and other letters shared the same locations in spatial frequency and in space, this seems less plausible. A further question, raised by a reviewer, is whether feature-based attention, as well as the spatial attention discussed here, may also suffer from focusing. As presented, the model does not distinguish between the two, and so it would imply that, if attention to featural noise exceeded attention to the target features, performance would suffer.

Since visual targets are often embedded in complex surrounds, rather than being isolated as is so common in laboratory experiments, these considerations from signal detection theory may have

implications for vision in natural scenes, and perhaps also for hearing in noisy environments. Signal detection theory does not predict that focusing attention *must* improve performance; this depends on the effects of attention on the signal and on the noise, both of which may be increased by attention, but unequally.

**Funding:** Data collection was supported by a grant from the William Randolph Hearst Foundation to Elizabeth Fine.

**Conflicts of Interest:** The authors declare no conflict of interest.

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
