# Peer review of "Attention and Signal Detection"

_information, doi:10.3390/info10080254_

Round 1

Reviewer 1 Report

The background on SDT is appreciated and I think sufficient for current purposes, and the general idea that attention is not always facilitatory I think is worth exploring.

Most of the major issues I have with this work is a lack of methodological and statistical surrounding the experiment (or at least lack of access to those details). I do not have access to the Fine & Reeves (2018) chapter, and I do not expect that interested readers would as well. As a result, crucial details are missing from the article.

1) Firstly, line 17 states that there are four placeholders (top,bottom,left,right from a central fixation). But Experiment 3 then introduces a 'diagonal format' (line 201). Are these just the four placeholders rotated by 45 degrees? Description would be clearer if there was a figure showing what the participant sees on-screen.

2) Secondly, how were participants identifying the letters in the non-binary case? Were they pressing a button or were they vocalizing it? Vocalizations add quite a bit of noise to reaction time data (differences in the way sounds are articulated, etc). If they were responding with the key of the letter presented, did they have to search a keyboard for the letter? That would add noise to the RT as well. In either case, the one effect size mentioned, 49ms slowing for the letter identification in the focused attention condition (line 195), was not particularly large. What is the measurement error of the technique used? What is the statistical significance of this result? What are the similar statistics for Experiment 2 and 3?

3) Another issue is whether the model accurately predicts the effect size of RTs changes. R does flip in direction when alpha (attention) is varied depending on the external noise (Ne), but what about the relative magnitude of those differences? Change in R as a function of s and alpha is much less when external noise is high, so one would expect small RT differences, comparing to when external noise is low. Is the magnitude of facilitatory effect of attention in the E forward/backward task much greater then the cost in the letter identification task? This is something that could explain the small effect size (49ms) of the latter. Also, what is the size of the RT benefit of attention for the E forward/backwards task?

4) Also, a major source of external noise is assumed to be stemming from the placeholders acting as masks (lines 221; 273 to 277). That being the case, would a direct test of the idea that high external noise is causing the lack of facilitation / cost of spatial attention not be to have a condition where there is no post-stimulus masking by the place holder (i.e. remove the placeholder)? It seems a control condition that is conspicuously absent.

5) More of a theoretical question: There seems to be mechanistic differences between feature and spatial attention (e.g. Carrasco, 2011 Vision Research). Do you expect the same hypothesis of external noise to affect feature based attention in a similar way, or is this something specific to spatial attention?

There are also a host of formatting issues. Not sure if this is due to the upload system or the source manuscript, but:

1) Missing line spacing after lines 338 and 340.

2) The way the equations are presented seems a bit strange. Are they supposed to be centered?

3) Citations have weird symbols (e.g. lines 219, 134)

4) Some of the spacings seem doubled. and some missing (lines 195, 54)

5) For line 98, "(Gilbert Ryle)" makes it seem like an incomplete citation. Consider rewording?

6) Concluding sentence seems weird as well ("I do not know") seems redundant as "perhaps" already conveys uncertainty. 

Author Response

The background on SDT is appreciated and I think sufficient for current purposes, and the general idea that attention is not always facilitatory I think is worth exploring.

Thank you!

Most of the major issues I have with this work is a lack of methodological and statistical surrounding the experiment (or at least lack of access to those details). I do not have access to the Fine & Reeves (2018) chapter, and I do not expect that interested readers would as well. As a result, crucial details are missing from the article.

I have tried to supply these without plagiarizing our earlier article.

1) Firstly, line 17 states that there are four placeholders (top,bottom,left,right from a central fixation). But Experiment 3 then introduces a 'diagonal format' (line 201). Are these just the four placeholders rotated by 45 degrees? Description would be clearer if there was a figure showing what the participant sees on-screen.

I think a verbal description is sufficient, but I have incorporated the suggested text, ‘the four placeholders rotated by 45 degrees’, as this is helpful. Thanks.

2) Secondly, how were participants identifying the letters in the non-binary case? Were they pressing a button or were they vocalizing it? Vocalizations add quite a bit of noise to reaction time data (differences in the way sounds are articulated, etc). If they were responding with the key of the letter presented, did they have to search a keyboard for the letter?

Good point. Thanks. The screen provided for the response is now described in the Method section, on page 8 of the ms word document

That would add noise to the RT as well. In either case, the one effect size mentioned, 49ms slowing for the letter identification in the focused attention condition (line 195), was not particularly large. What is the measurement error of the technique used? What is the statistical significance of this result? What are the similar statistics for Experiment 2 and 3?

All statistics included. Remark: 49 ms is much larger than the error of measurement (8 ms,); indeed, in many cognitive papers, a difference of even 10ms would be of interest. The decisions are rapid, so 49ms is not trivial in comparison to the overall mean (489 ms. )

3) Another issue is whether the model accurately predicts the effect size of RTs changes.

It does, but at the cost of employing two free parameters, the scale factor b to convert R to RT, and the ‘simple’ or motor component, RTo. This is a standard problem with RTs; only in the case of much slower responses, taking many seconds, can RT0 be ignored. The fit is now indicated on the graph (Fig. 1). I had not wanted to portray this, as the basic issue is not the fit (the primary interest for many researchers) but rather the logic – that signals can be detected worse when fully attended – but I do see the reviewer’s point and I have acceded to it.

R does flip in direction when alpha (attention) is varied depending on the external noise (Ne), but what about the relative magnitude of those differences? Change in R as a function of s and alpha is much less when external noise is high, so one would expect small RT differences, comparing to when external noise is low. Is the magnitude of facilitatory effect of attention in the E forward/backward task much greater than the cost in the letter identification task? This is something that could explain the small effect size (49ms) of the latter. Also, what is the size of the RT benefit of attention for the E forward/backwards task?

Good point. The data upon which the model is fit are now plotted on the same graph, to permit comparison. It is a weakness of the current study that we did not vary S (stimulus strength) parametrically; had we done so, we could have established whether the ‘flip’ from sots with letters to benefits with E’s happened at other performance levels, too.

4) Also, a major source of external noise is assumed to be stemming from the placeholders acting as masks (lines 221; 273 to 277). That being the case, would a direct test of the idea that high external noise is causing the lack of facilitation / cost of spatial attention not be to have a condition where there is no post-stimulus masking by the place holder (i.e. remove the placeholder)? It seems a control condition that is conspicuously absent.

Correct. This would have been a nice experiment, to compare the forward and backward masking effects of the placeholders, to establish which is dominant. The conclusion now includes the statement: An experiment in which the placeholders did not return, so that only forward masking occurred, could test this notion; whereas patterned backward masks limit processing time, forward masks do not (Liss, 1968; Navon & Purcell, 1981).

5) More of a theoretical question: There seems to be mechanistic differences between feature and spatial attention (e.g. Carrasco, 2011 Vision Research). Do you expect the same hypothesis of external noise to affect feature based attention in a similar way, or is this something specific to spatial attention?

What an interesting question. I have no idea. The way that the argument is presented, feature-based attention could also cause a performance loss, if attending to the features also increased the noise. In that case, the noise would be systematic and presumably not Gaussian, but I don’t see any real problem here – a multinomial distribution, for example, might work well enough. At any events, I hope the reviewer doesn’t mind if I incorporate his suggestion in the Discussion, as follows: “As suggested by a reviewer, given the present argument, feature-based attention could also cause a performance loss, if attending to the features also increased the noise more than the signal. However I know of no relevant data to test this prediction”.

There are also a host of formatting issues. Not sure if this is due to the upload system or the source manuscript, but:

I have tried to fix these in both versions of the ms, and made other small corrections as well.

1) Missing line spacing after lines 338 and 340. Done.

2) The way the equations are presented seems a bit strange. Are they supposed to be centered?

They hug the left-hand margin, but I will center them if requested.

3) Citations have weird symbols (e.g. lines 219, 134). fixed

4) Some of the spacings seem doubled. and some missing (lines 195, 54) seems OK in new pdf.

5) For line 98, "(Gilbert Ryle)" makes it seem like an incomplete citation. Consider rewording?

Removed. Ryle is not really relevant here.

6) Concluding sentence seems weird as well ("I do not know") seems redundant as "perhaps" already conveys uncertainty.

Change made.

Reviewer 2 Report

The researcher sought to use Signal Detection Theory to explain the way in which focusing attention may not necessarily lead to improved performance; the case study used was a prior experiment of Fine and Reeves (2018) in which the focusing of attention improved performance on the vertical meridian when the target was known, but impeded performance on that meridian when the target was unknown. There is some concern that other interpretations of the results were not taken into account. There is also some concern that the manuscript lacks value as it reads more as a Discussion section of a Research paper without really adding novelty; indeed, the concept that there are instances in which attention can impede performance is not new, as will be detailed below.  

Below are some concerns, and suggestions for clarification, in no particular order.  

p.1, line 30, It is noted that there is a “sense in which attention implies conscious awareness”. There is no reference to that allusion. In fact, there are papers to suggest that attention can occur non-consciously (e.g., Ansorge & Heumann, 2006; Sato, Okada, Toichi, 2007; Gayet,  Van der Stigchel & Paffen, 2013; for a review Mulckhuyse & Theeuwes, 2010). I recommend that the researcher further clarifies their position on this phenomenon.  

p. 1, line 34, It is noted that items selected by attention will be processed “better” and rejected items “worse”. It is unclear – even given the provided reference – what evidence exists to support the concept that attention affects the quality of information processing. Indeed, attention refers to a process that enables information selection or filtering by affecting the depth of information processing (e.g., Bentin, 1998; Craik & Lockhart, 1972).  I recommend that the researcher elaborates their reasoning for such conceptualisation.

p.3, line 106, The researcher uses signal detection theory to inform different “aspect[s] of attention” without first clarifying the definition of attention that they have implemented throughout the manuscript. Indeed, it is implied that the “aspect[s] of attention” (selection through gating, creating sets, etc.) are distinct and exclusive from each other, which is highlighted by the usage of attention as a property that you can increase or decrease (e.g., p.2, line 86, “paying attention”). There is behavioural and neural evidence to suggest that attention is not a property in and of itself, but a multilevel process (Kastner & Pinsk, 2004). I recommend that the researcher provides an evidence-based definition of attention.   

Italicise d’ throughout the paper, as per common convention.

p.2, line 92, I recommend providing an information selection rather than social example (i.e., crime) to explain the way in which signal detection theory can operate in the real world.

p.3, line 107, It is noted that increasing stimulus exposure time leads to increased resources, which is synonymised with increased attention. No reference is provided for such a supposition. First, increasing stimulus exposure time may increase the depth of information processing, but that need not mean increased attentional or cognitive resources. Second, the reason that “increasing resources” is equated with “increasing attention” is unclear. Again, as per my comment in (3), attention is a multi-level process, which should not be reduced to merely resource allocation. I recommend the researcher clarifies these aspects.

p. 3, line 142, It is stated here that “attention must improve performance by filtering out noise or decreasing uncertainty about the signal or directly enhancing the signal”, see Serences (2011) for alternate (or additional) explanations of the effect of attention. Please include a discussion of these explanations.

p. 4, line 174, It was stated that “attention is by definition facilitatory”, but this statement negates inhibition of return. For what reason was this phenomenon not considered?

p.4, line 184, For what reason was backward masking employed in this task? Was the purpose to examine attentional orienting under non-conscious conditions?

p. 4, In explaining the Fine and Reeves (2018) findings, there are other, non-attentional explanations that should be taken into account: (1) different time-courses of facilitation and inhibition of return for known (small set size) vs. unknown (large set size) targets or targets that require multiple feature integration versus singular feature search; (2) differential attentional allocation for different target types when attention is initially deployed globally versus focally (Kubovy et al., 1999). Indeed, in both (1) and (2) target expectancy in the aforementioned experiment is confounded by the level of feature integration, which is higher when identifying one of 20 letters, and lower when identifying forward/backward Es; and (3) amplification of the horizontal-vertical anisotropy in visual attention (e.g., Al-Janabi & Greenberg, 2016; Barnas & Greenberg, 2016) for different target types and/or different target expectancies when attention is initially deployed globally versus focally. These explanations, in addition to the SDT, should also be taken into account to explain findings.

p. 6. line 274, It is stated that “If the gate must be held open for longer to identify individual letters than to detect the orientation of the E’s, external noise from the placeholders (acting as masks) will be greater in the former than the latter task”; this explanation is not novel. First, it is known that the presentation of a forward (but not backward) mask limits the processing of the preceding stimulus, at times rendering that stimulus non-conscious (e.g., Kouider, 2007). Second, as mentioned in (2) above, it has been shown that the time-course of feature integration vs. single feature search differ (e.g., Decco et al., 2002). Both points here put into question the novelty and value of this manuscript. The researcher ought to better motivate and place this paper into the broader context in order for its value to be discerned.   

Author Response

 Reviewer 2.

The researcher sought to use Signal Detection Theory to explain the way in which focusing attention may not necessarily lead to improved performance; the case study used was a prior experiment of Fine and Reeves (2018) in which the focusing of attention improved performance on the vertical meridian when the target was known, but impeded performance on that meridian when the target was unknown. There is some concern that other interpretations of the results were not taken into account. There is also some concern that the manuscript lacks value as it reads more as a Discussion section of a Research paper without really adding novelty; indeed, the concept that there are instances in which attention can impede performance is not new, as will be detailed below.

Below are some concerns, and suggestions for clarification, in no particular order.

p.1, line 30, It is noted that there is a “sense in which attention implies conscious awareness”. There is no reference to that allusion. In fact, there are papers to suggest that attention can occur non-consciously (e.g., Ansorge & Heumann, 2006; Sato, Okada, Toichi, 2007; Gayet, Van der Stigchel & Paffen, 2013; for a review Mulckhuyse & Theeuwes, 2010). I recommend that the researcher further clarifies their position on this phenomenon.

I agree 100% with the reviewer; my wording was poor. The implication refers to the phenomenology, that is, to the extent that people are aware of paying more (or less) attention. Fixed.

p. 1, line 34, It is noted that items selected by attention will be processed “better” and rejected items “worse”. It is unclear – even given the provided reference – what evidence exists to support the concept that attention affects the quality of information processing. Indeed, attention refers to a process that enables information selection or filtering by affecting the depth of information processing (e.g., Bentin, 1998; Craik & Lockhart, 1972). I recommend that the researcher elaborates their reasoning for such conceptualisation.

None of the papers referenced prove that pure selection occurs, i.e. that unattended items are totally lost. As first shown by Anne Treisman, in Donald Broadbent’s lab, attention attenuates, it does not select in an all-or-none fashion; Broadbent himself told me (when a graduate student) that his ‘filter’ model was rejected by her data. The term ‘better’ surely includes an all-or-none possibility, a graded possibility, and a depth-or-processing possibility, and is chosen to be generic, as the model doesn’t specify the mechanism other than in terms of S/N ratio. So, I keep it. The Craik & Lockhart, 1972, reference is interesting. I suspect that the depth of processing for E’s and letters was in both cases shallow, as the letters were not parts of words but were distinct, more like stand-alone shapes. As I have no evidence either way, I have not discussed this issue.

p.3, line 106, The researcher uses signal detection theory to inform different “aspect[s] of attention” without first clarifying the definition of attention that they have implemented throughout the manuscript.

OK, fixed. Good point; needs to be done early. I have inserted a heading: the role of selective attention, in the Introduction, to draw the reader’s eye towards the definition I employ.

Indeed, it is implied that the “aspect[s] of attention” (selection through gating, creating sets, etc.) are distinct and exclusive from each other, which is highlighted by the usage of attention as a property that you can increase or decrease (e.g., p.2, line 86, “paying attention”). There is behavioural and neural evidence to suggest that attention is not a property in and of itself, but a multilevel process (Kastner & Pinsk, 2004). I recommend that the researcher provides an evidence-based definition of attention.

We listed 20 different ways in which attention has been operationalized in the book chapter, along with the different definitions and experiments, associated with each. Here we discuss just the one way in which it can relate to our experiment, namely, operationalized as focusing/diffusing. The reviewer is quite correct that this has to be made explicit, early on. Thanks!

Italicise d’ throughout the paper, as per common convention.

p.2, line 92, I recommend providing an information selection rather than social example (i.e., crime) to explain the way in which signal detection theory can operate in the real world.

Crime has the great advantage that the separation between sensor (police) and decision (jury, judge) is made rigid in law, to prevent the decision-maker from influencing the sensor. As such it provides a vivid and easily-understood metaphor. So I have kept it. If either the Editor or reviewer can provide a useful ‘information selection’ metaphor, I will be happy to use it.

p.3, line 107, It is noted that increasing stimulus exposure time leads to increased resources, which is synonymised with increased attention. No reference is provided for such a supposition.

Good point. Reference now given (M. Carrasco et al, 2002). I had taken this for granted, not imagining that anyone would object! ). But I actually meant that increasing attention will increase the evidence; I did not equate them, as there are other ways of increasing the evidence, for example, by training.

First, increasing stimulus exposure time may increase the depth of information processing, but that need not mean increased attentional or cognitive resources. Second, the reason that “increasing resources” is equated with “increasing attention” is unclear. Again, as per my comment in (3), attention is a multi-level process, which should not be reduced to merely resource allocation. I recommend the researcher clarifies these aspects.

But the reviewer is wrong about the logic here. Signal detection theory requires only that there is a dimension of activity, from less to more, that corresponds to ‘strength’, and the role of attention is just to modulate this. The issue about stages of processing of the stimulus, like the issue of levels or modes of attention, is not relevant to the formalism. SDT simply doesn’t care why the evidence for a stimulus is stronger or weaker, or what ingredients had to be cooked up to make it so; it only cares that the outcome can be represented on a single dimension. Analogy; Numerous levels of the criminal justice system may be involved in assembling the evidence against a person, but, like the complexities of how attention influences each level, this is not relevant; only the final evidence before the court, strong or weak, is relevant.

p. 3, line 142, It is stated here that “attention must improve performance by filtering out noise or decreasing uncertainty about the signal or directly enhancing the signal”, see Serences (2011) for alternate (or additional) explanations of the effect of attention. Please include a discussion of these explanations.

I meant, in the context of signal detection theory. Now specified. (There are many other ways in which attention could influence performance, but our experiment was designed to exclude them.) Thanks for noticing this slip in my exposition.

p. 4, line 174, It was stated that “attention is by definition facilitatory”, but this statement negates inhibition of return. For what reason was this phenomenon not considered?

Interesting ! IOR refers to a rebound following the concentration of attention, the latter being facilitatory. A rebound later on may well be inhibitory. Our experiment did involve focusing, i.e. concentration of attention, but it did not involve IOR type rebounds as published in the literature- the time scale is much too short. So I don’t see how the idea could be applied. One would have to assume fast rebounds for letters, within 50ms, but no rebounds for E’s; and why would that be? I am willing to entertain this possibility in a footnote, if the Editor advises it, but I cannot see speculating about it in the text, as it seems like too much of a ‘reach’.

p.4, line 184, For what reason was backward masking employed in this task? Was the purpose to examine attentional orienting under non-conscious conditions?

No. The purpose was to limit exposure to the target letter or E, so that performance would not be at ceiling. In no case was the mask used to prevent awareness. This point is now made in the Method, so that readers are not mislead by the ‘masking’ terminology. Lowering the presentation duration to the point that the letters or E’s were not noticed by the subject could be an interesting experiment to do, as it could address the disposition of unconscious attention, but alas, we did not do it.

p. 4, In explaining the Fine and Reeves (2018) findings, there are other, non-attentional explanations that should be taken into account: (1) different time-courses of facilitation and inhibition of return for known (small set size) vs. unknown (large set size) targets or targets that require multiple feature integration versus singular feature search;

IOR does not occur in the time period under consideration.

(2) differential attentional allocation for different target types when attention is initially deployed globally versus focally (Kubovy et al., 1999).

Yes; in fact, our results with E’s and letters go along with this general point. Global (diffuse) attention helps letters but hinders E’s. Just the strange result that the SDT model helps explain.

The cited reference appears to be Feature integration that routinely occurs without focal attention by M Kubovy, DJ Cohen, J Hollier - Psychonomic Bulletin & Review, 1999. They showed that ‘the subsystems that process form and color subsystems interact during the preattentive processing of feature-dependent information.’ Kubovy’s other 1999 publications refer to gabor lattices and also don’t appear relevant. I will be happy to cite Kubovy if I can do so correctly.

Indeed, in both (1) and (2) target expectancy in the aforementioned experiment is confounded by the level of feature integration, which is higher when identifying one of 20 letters, and lower when identifying forward/backward Es;

Yes, as stated, the letters have to be processed further, to be identified, than the E’s, which act as optotypes. This explains the longer SOA needed for the letters, and, in the model, accounts for the opposite effect on letters (because the attention-grabbing response includes more ‘noise’ from the placeholders). However I have no reason to think that ‘feature integration’ is relevant; the letters are well known to readers, and act like templates. The idea of feature integration is that different features, say, color and form, need on-line integration to be bound. So I appreciate this idea, as a suggestion, but I regard it as tangential to the paper and do not discuss it.

and (3) amplification of the horizontal-vertical anisotropy in visual attention (e.g., Al-Janabi & Greenberg, 2016; Barnas & Greenberg, 2016) for different target types and/or different target expectancies when attention is initially deployed globally versus focally. These explanations, in addition to the SDT, should also be taken into account to explain findings.

Thanks for this reference! These authors found that “revealed a horizontal shift advantage (faster RTs for horizontal shifts across the vertical meridian compared to vertical shifts across the horizontal meridian),” which is consistent with our RTs, although in a different domain (object-based attention) .

Barnas, A. J., & Greenberg, A. S. (2016). Visual field meridians modulate the reallocation of object-based attention. Attention, Perception, & Psychophysics, 78, 1985-1997. doi:10.3758/s13414-016-1116-5

p. 6. line 274, It is stated that “If the gate must be held open for longer to identify individual letters than to detect the orientation of the E’s, external noise from the placeholders (acting as masks) will be greater in the former than the latter task”; this explanation is not novel.

First, it is known that the presentation of a forward (but not backward) mask limits the processing of the preceding stimulus, at times rendering that stimulus non-conscious (e.g., Kouider, 2007).

Liss (1968), like many others at that time, found the opposite; the backward, not the forward, mask limited persistence and in doing so, ‘stopped processing’ of the target. (Also, none of our stimuli in that research or the present ms were non-conscious.) Reference now given.

Second, as mentioned in (2) above, it has been shown that the time-course of feature integration vs. single feature search differ (e.g., Decco et al., 2002). Both points here put into question the novelty and value of this manuscript. The researcher ought to better motivate and place this paper into the broader context in order for its value to be discerned.

I am OK that the reviewer thinks the explanation in terms of the gate allowing in external noise is not novel. But it was not presented for its novelty or otherwise. It was presented as a rational way to explain why focusing attention could be deleterious, given the gate duration measured by Reeves & Sperling. The point of the present paper is to point out that attention may be deleterious even when signal detection is optimal, and give a reason why; it is not to give a historical review, or to claim that no other account is possible. Moreover, as stated above, I believe the reviewer is incorrect in assuming that the letters required ‘feature integration’, as usually understood, whereas the E’s did not. I am willing to present this idea in a footnote, and attribute it to the reviewer, with the disclaimer that the author does not share this view, if the Editor requests this.

Reviewer 3 Report

The paper is in general very well written with a straightforward agenda: Point out, that the consequences of attention in signal-detection accounts can be counter-intuitive, with attention having a negative impact on performance. Results from experiments are described, in some of which observers performed better when attention was diffused over more locations compared to conditions where attention was distributed to less elements. The result is linked to the differntial enhancement of external and internal noise.

I have only a few points (listed in no order of importance):

(1) I enjoyed that the introduction provides recap of SDT and the role of atttention which is detailed yet concise. While everything is explained in small steps, z-transformations are used without introducing them or even mentioning what "z( )" is. I suggest to spend half a sentence or so to point out what that is ...

(2) The text would really benefit from proper use of math font

(3) Some parts appear unfinished, with placeholders as references "(Pelli, 198*), (Lu & Dosher, 200*), (Carrasco, 20**). The references are missing.

(4) The paper is written without any subsections (except "Introduction" - everything is the introduction :-) ). In general I like that the relatively short text is presented as one coherent thought. However, especially where the experiments are reported, a bit more structuring would be nice. E.g.: "Our subjects (a total of 78 young adults, all with 20:20 vision or better)" -- is that for all experiments taken together? Are all results discussed from Fine & Reeves, 2018?

(5) The overlap of this paper and Fine & Reeves (2018) is unclear to me (I was not able to obtain the text). From the abstract I expected original experiments reported in here. After reading the paper and rereading the abstract I see that that could refer to results of earlier studies. I have no problem with any overlap; indeed, the current publication might make the results better accessible. But of course that's for the journal to decide if the current article is sufficient original (if reviewers should comment on that, Fine & Reeves (2018) [and maybe others?] should be provided as a reviewer supplement).

I would appreciate however if the author makes it more explicit in the text that earlier data is discussed and what the added value from the new manuscript is.

(6) As for the differences in the results with optotype types and letters I was wondering whether confusability to the elements and the different target set sizes might play a role. Process-based models of sitimulus encoding might be illuminating here. For instance, Bundesen's (1990) theory should have something to say about this. If I remember correctly, Bundesen & Habekost (2008) also relate the model to STD.

Author Response

Reviewer 3

The paper is in general very well written with a straightforward agenda: Point out, that the consequences of attention in signal-detection accounts can be counter-intuitive, with attention having a negative impact on performance. Results from experiments are described, in some of which observers performed better when attention was diffused over more locations compared to conditions where attention was distributed to less elements. The result is linked to the differntial enhancement of external and internal noise. I have only a few points (listed in no order of importance):

(1) I enjoyed that the introduction provides recap of SDT and the role of atttention which is detailed yet concise. While everything is explained in small steps, z-transformations are used without introducing them or even mentioning what "z( )" is. I suggest to spend half a sentence or so to point out what that is ...

done

(2) The text would really benefit from proper use of math font

It is, in the word version.

(3) Some parts appear unfinished, with placeholders as references "(Pelli, 198*), (Lu & Dosher, 200*), (Carrasco, 20**). The references are missing.

Fixed. My apologies for this oversight.

(4) The paper is written without any subsections (except "Introduction" - everything is the introduction :-) ). In general I like that the relatively short text is presented as one coherent thought. However, especially where the experiments are reported, a bit more structuring would be nice. E.g.: "Our subjects (a total of 78 young adults, all with 20:20 vision or better)" -- is that for all experiments taken together? Are all results discussed from Fine & Reeves, 2018?

done

(5) The overlap of this paper and Fine & Reeves (2018) is unclear to me (I was not able to obtain the text). From the abstract I expected original experiments reported in here. After reading the paper and rereading the abstract I see that that could refer to results of earlier studies. I have no problem with any overlap; indeed, the current publication might make the results better accessible. But of course that's for the journal to decide if the current article is sufficient original (if reviewers should comment on that, Fine & Reeves (2018) [and maybe others?] should be provided as a reviewer supplement).

I would appreciate however if the author makes it more explicit in the text that earlier data is discussed and what the added value from the new manuscript is.

Now clarified at the start of METHOD. The model is the focus of this paper. The data are selected from those referred to in the chapter, which goes into more conditions.

(6) As for the differences in the results with optotype types and letters I was wondering whether confusability to the elements and the different target set sizes might play a role. Process-based models of stimulus encoding might be illuminating here. For instance, Bundesen's (1990) theory should have something to say about this. If I remember correctly, Bundesen & Habekost (2008) also relate the model to STD.

Yes! We also had wondered about this, and ran a control (Experiment 4) in which letters were sorted into two groups, 5 vowels and 5 consonants, so that the response could be binary. The results were the same as with the letters. I now refer to this result, without going into details..

Round 2

Reviewer 1 Report

Thank you for addressing my earlier concerns and for incorporating some of the suggestions. There are a couple of minor formatting issues listed below that could be fixed, otherwise I see no further need for improvements/review.

1) Line 271, author names are underlined.

2) Figure 1, spacing in labels are slightly inconsistent (e.g. Mon2, vs Mon 2).

3) Also Figure 1, maybe shift the y-axis label (response strength) more to the left? It is overlapping with the axis values.

4) In the references, some random underlines and bold text, e.g. citation 5,12,20,23

Author Response

1) Line 271, author names are underlined.

2) Figure 1, spacing in labels are slightly inconsistent (e.g. Mon2, vs Mon 2).

3) Also Figure 1, maybe shift the y-axis label (response strength) more to the left? It is overlapping with the axis values.

4) In the references, some random underlines and bold text, e.g. citation 5,12,20,23

I have tried to remove the underlines, which were introduced in the proof. Ref. 23 still has a green underline I cannot remove.

The original for fig. 1 is properly spaced. I am uploading it again as I cannot edit the figure in the proof.

Thankyou for your careful eye !

adam reeves

Reviewer 2 Report

The reviewer is satisfied with the amendments made by the author, and thanks them for the invigorating discussion on the comments. I make only one further suggestion, where the author states "attention by definition is facilitatory", it would be wise to clarify that attention is only such under the currently examined conditions (because of the used exposure durations). As the author agreed, attention can at times be inhibitory.

Author Response

The reviewer is satisfied with the amendments made by the author, and thanks them for the invigorating discussion on the comments. I make only one further suggestion, where the author states "attention by definition is facilitatory", it would be wise to clarify that attention is only such under the currently examined conditions (because of the used exposure durations). As the author agreed, attention can at times be inhibitory.

good point; change made in line 248.

One simple notion is that, in the current conditions, attention always improves processing, Thanks.